# In Vitro Cytotoxic and Genotoxic Evaluation of Nitazenes, a Potent Class of New Synthetic Opioids

**DOI:** 10.3390/jox15060203

**Published:** 2025-12-02

**Authors:** Francesca Rombolà, Sara Bartoletti, Sabrine Bilel, Patrizia Hrelia, Matteo Marti, Monia Lenzi

**Affiliations:** 1Department of Pharmacy and Biotechnology, Alma Mater Studiorum University of Bologna, 40126 Bologna, Italy; francesca.rombola3@unibo.it (F.R.); sara.bartoletti3@studio.unibo.it (S.B.); patrizia.hrelia@unibo.it (P.H.); 2Department of Translational Medicine, Section of Legal Medicine, LTTA Center and University Center of Gender Medicine, University of Ferrara, 44121 Ferrara, Italy; sabrine.bilel@unife.it (S.B.); matteo.marti@unife.it (M.M.); 3Clinical Pharmacology Unit, IRCCS AOU of Bologna, 40138 Bologna, Italy; 4Collaborative Center for the Italian National Early Warning System (NEWS-D), Department of Anti-Drug Policies and Other Addictions, Presidency of the Council of Ministers, 00186 Rome, Italy

**Keywords:** nitazenes, new synthetic opioids, novel psychoactive substances, genotoxicity, in vitro mammalian cell micronucleus test, flow cytometry, TK6 cells

## Abstract

In recent years, the expansion of the illicit market for Novel Psychoactive Substances (NPS) has resulted in the emergence of numerous synthetic recreational drugs specifically designed to evade legal control and analytical detection. Among these, nitazenes represent one of the most potent classes of new synthetic opioids, although information regarding their toxicological properties remains limited. The present study aimed to assess the genotoxic potential of four nitazenes: clonitazene, etonitazene, isotonitazene and metonitazene in human lymphoblastoid TK6 cells using a flow cytometric version of the In Vitro Mammalian Cell Micronucleus Test, following OECD Guideline No. 487. Cells were exposed to concentrations ranging from 12.5 to 100 μM, and cytotoxicity, cytostasis, and apoptosis were evaluated to identify appropriate doses for micronucleus frequency assessment. Vinblastine, a well-established mutagen, was included as positive control. Our findings demonstrated that clonitazene and isotonitazene exhibit mutagenic potential, suggesting an increased long-term risk of developing chronic degenerative diseases. Furthermore, the results revealed that structurally related molecules can induce markedly different cellular effects, underscoring the importance of compound-specific toxicological evaluations to achieve a comprehensive understanding of the risks associated with their illicit use—risks often presumed to involve only addiction or acute toxicity.

## 1. Introduction

Over the past decade, the availability and use of New Psychoactive Substances (NPS), a class of synthetic recreational drugs designed to evade legal classification and standard detection methods, have increased significantly [1]. Among these, New Synthetic Opioids (NSOs) have emerged as the fastest-growing and most dangerous subgroup, characterized by extreme potency and strong association with severe intoxications and fatalities [2,3].

The increased availability of NSOs on the illicit market has created a real “opioid crisis”, particularly affecting North America, Oceania, and Western and in Central Europe [4].

Currently, the European Union Drugs Agency (EUDA) is monitoring 81 new opioids and in 2024 the highest number of these compounds was reported to the EU Early Warning System (EWS) [5].

The most recent NSO subclass to emerge is the 2-benzylbenzimidazoles opioids, commonly known as ‘nitazenes’, marked by a potency that can exceed both morphine and fentanyl. These compounds were first synthesized in 1957 by the Swiss pharmaceutical company Chemische Industrie Basel (CIBA) with the aim of developing new analgesics that would be more potent than morphine while producing fewer side effects. However, they were never commercialized due to their high risk of adverse effects, including addiction and overdose [3,5]. Since 2019, nitazenes have emerged on recreational drug markets in the form of powders, counterfeit tablets, and liquids, and have been identified in illicit heroin, fentanyl, as well as in adulterated benzodiazepines and analgesics, thereby further exacerbating the opioid crisis [6]. By the end of 2024, EUDA was officially monitoring a total of 22 nitazenes, 7 of which had been formally reported for the first time during that year [1].

Nitazenes are a subclass of non-fentanyl analogues NSOs that, despite a chemical structure distinct from classical μ-opioid receptor agonists, retain the ability to bind these receptors and elicit typical opioids effects, such as sedation, euphoria and respiratory depression that can progress to death [7].

Isotonitazene was among the first nitazenes associated with fatalities in Europe in 2019 [2,8]. In response to the rapidly rising number of intoxications and fatalities linked to isotonitazene, awareness of the substance increased, and various national and international control measures were subsequently announced and implemented the U.S. Drug Enforcement Administration (DEA) has classified metonitazene and isotonitazene (both classified between 2021 and 2024) [1]. Many member states have introduced laws that further restrict nitazenes beyond the 1961 Convention. Some countries have specific regulations for nitazenes, while others rely on broader opioid control laws [2]. This variability poses challenges for law enforcement and public health responses to the emerging threat of nitazenes. Nevertheless, limited information is available regarding the effects of these compounds on humans and, in particular, studies on the genotoxicity of nitazenes are not yet available [2,9]. However, other NSOs, including fentanyl analogues such as Acrylfentanyl, Ocfentanyl and Furanylfentanyl [10] and non-fentanyl analogues such as brorphine [11] and related compounds, were found to be genotoxic, which supports the importance of evaluating this aspect also for the highly potent nitazene opioids.

For this reason, the present study aims to evaluate the genotoxicity of the four classic nitazenes (clonitazene, etonitazene, isotonitazene and metonitazene) (Figure 1) in human TK6 lymphoblastoid cells, in terms of their capacity to lead to structural and numerical chromosomal aberrations.

For this purpose, the “In Vitro Mammalian Cell Micronucleus Test” was performed in accordance with OECD Guideline No. 487 [12]. In the initial phase, TK6 cells were exposed to the compounds under investigation at concentrations ranging from 12.5 to 100 µM. Cytotoxicity and cytostasis were subsequently evaluated to identify concentrations that maintained cell viability and proliferation at ≥45% ± 5%. When cytotoxicity and cytostasis exceeds this threshold the formation of micronuclei is not possible. Apoptosis was also assessed to ensure that its induction remained below a twofold increase relative to the negative control, as recommended by the guideline.

Accordingly, the concentrations appropriate for the assessment of micronuclei (MNi) frequency were selected, and the analysis was performed using a flow cytometry (FCM) protocol previously developed and validated in our laboratory [13].

## 2. Materials and Methods

### 2.1. Reagents

The following were used: dimethyl sulfoxide (DMSO), ethylenediaminetetraacetic acid (EDTA), fetal bovine serum (FBS), Annexin V-Alexa Fluor 488, Annexin-binding buffer, L-Glutamine (L-GLU), Nonidet, Penicillin-Streptomycin solution (PS), Phosphate-Buffered Saline (PBS), Potassium Chloride, Potassium Dihydrogen Phosphate, Propidium Iodide (PI), Roswell Park Memorial Institute (RPMI) 1640 medium, BPC-grade water, Sodium Chloride, Sodium Hydrogen Phosphate, Vinblastine (Vinb) (all purchased from Merck, Darmstadt, Germany), RNase A, SYTOX Green (purchased from Thermo Fisher Scientific, Waltham, MA, USA).

### 2.2. Nitazenes

The four nitazenes under investigation (clonitazene, etonitazene, isotonitazene and metonitazene) were purchased from LGC Standards S.r.L. (Milan, Italy) and stored at −20 °C. At the time of use they were dissolved in absolute ethanol at a maximum concentration of 10 mM.

### 2.3. Cell Culture

All experiments were performed using the human lymphoblastoid cell line TK6, validated by OECD guidelines as suitable for micronucleus (MN) detection [9,14]. TK6 cells were obtained from ATCC (Manassas, VA, USA) and cultured at 37 °C in a humidified atmosphere with 5% CO_2_ in RPMI-1640 medium supplemented with 10% fetal bovine serum (FBS), 1% L-glutamine, and 1% penicillin–streptomycin (PS). Considering the approximate doubling time of 13 h, cultures were diluted with fresh medium every two days to maintain exponential growth, ensuring that cell density did not exceed 1 × 10^6^ cells/mL.

### 2.4. Test Condition

TK6 cells were treated with the tested substances for 26 h. This time frame corresponds to approximately two cell cycles of TK6 cells, as indicated in OECD Guideline No. 487 [12].

#### 2.4.1. Selection of Concentration

The concentrations tested were initially selected in the range of 12.5–100 µM starting from the maximum solubility in ethanol (10 mM) and ensuring that the volume of SKS added never exceeded 1% *v*/*v* to avoid potential solvent toxicity for cells.

For the assessment of MNi frequency, concentrations were subsequently selected according to OECD Guideline No. 487, which establishes a cytotoxicity and cytostasis threshold of 55 ± 5% relative to the negative control. Therefore, only concentrations that maintained cell viability and proliferation at or above 45 ± 5% were tested [12].

#### 2.4.2. Measurement of Cytotoxicity

To evaluate cytotoxicity, aliquots of 2.5 × 10^5^ TK6 cells were seeded and subsequently treated with clonitazene, etonitazene, isotonitazene and metonitazene in a concentration range of 12.5–100 µM for 26 h. At the end of the treatments, cytotoxicity was assessed by determining the percentage of live cells using the propidium iodide (PI) test: the cells were stained with this reagent, which distinguishes live cells from necrotic cells by emitting red fluorescence. Cell viability was calculated automatically using Guava guavaSoft™ 4.5 software (Cytek Biosciences, Inc., Fremont, CA, USA) by acquiring 1000 events (cells) for each sample. Subsequently, the percentage of viability obtained in each sample was normalized with respect to that of the control cultures set at 100% [13].

#### 2.4.3. Measurement of Cytostasis

Aliquots of 2.5 × 10^5^ TK6 cells were treated with clonitazene, etonitazene, isotonitazene and metonitazene in the concentration range of 12.5–100 µM for 26 h. Cytostasis was evaluated using the PI assay and calculated automatically using Guava software guavaSoft™ 4.5 (Cytek Biosciences, Inc., Fremont, CA, USA). Specifically, correct cell replication was verified by comparing the number of cells seeded at time zero with the number of cells measured at the end of the treatment period [13]. In this way, Population Doubling (PD) was calculated (Equation (1)):(1)$$PD=log\left(\frac{post-treatment\;cell\;number}{initial\;cell\;number}\right)÷log2$$

After, to verify that most cells had completed cell division after treatment, the Relative Population Doubling (RPD) was calculated by comparing the PD values obtained in the negative controls with those of the treated cultures (Equation (2)).(2)$$RPD=\frac{PD\;in\;treated\;coltures}{PD\;in\;control\;coltures}\times 100$$

#### 2.4.4. Measurement of Apoptosis

To select the concentrations for the MNi frequency analysis, the potential activation of apoptosis as an alternative cell death pathway to necrosis was also assessed. Concentrations that induced a two-fold increase in apoptosis compared to the negative control were excluded. Based on cytotoxicity and cytostasis data, aliquots of 2.5 × 10^5^ TK6 cells were treated with the selected concentrations (12.5–100 µM) for 26 h. At the end of the treatments, cells were stained with two fluorophores: PI, which discriminates viable from necrotic cells by emitting red fluorescence, and Annexin V Alexa Fluor 488, which detects apoptotic cells by emitting yellow fluorescence. Double staining with Annexin V Alexa Fluor 488 and PI was performed to distinguish viable, early apoptotic, and late apoptotic/necrotic cells. Live cells were identified as PI−Annexin−, early apoptotic cells as PI−/Annexin V+, and late apoptotic or necrotic cells as PI+/Annexin V+.

The percentages of viable, early apoptotic, and necrotic cells were automatically calculated using GuavaSoft™ 4.5 software (Cytek Biosciences, Inc., Fremont, CA, USA) from 2000 acquired events. The percentage of apoptotic cells in treated samples was then normalized to that of the negative control (set to 1) and expressed as fold increase in apoptosis [12].

#### 2.4.5. Measurement of MNi Frequency

Aliquots of 2.5 × 10^5^ TK6 cells were treated with different concentrations of test substances selected based on cytotoxicity, cytostasis and apoptosis results. Specifically, clonitazene was tested at concentrations of 25 and 50 µM, etonitazene at 12.5 and 25 µM, isotonitazene at 50 and 75 µM, metonitazene at 25 and 50 µM.

In addition, the agent Vinblastine (Vinb), a known mutagenic agent, was used as positive control.

At the end of treatment, the MNi frequency was assessed using a flow cytometry protocol developed and published by Lenzi et al. [13]. Specifically, aliquots of 5 × 10^5^ cells treated with the selected concentration of the four nitazenes (Table 1) were collected and incubated with a lysis solution containing the fluorophore SYTOX Green, which enables the identification of nuclei and MNi by their green fluorescence.

The MNi frequency was calculated as the number of MNi per 5000 nuclei derived from viable and proliferating cells. Subsequently, the MNi frequency recorded in the treated cultures was normalized with respect to that recorded in the negative control cultures, set equal to 1, and expressed as an increase in MNi frequency.

#### 2.4.6. Flow Cytometry

FCM analyses were carried out with a Guava EasyCyte 5HT Flow Cytometer II generation system equipped with a class IIIb laser operating at 488 nm (Cytek Biosciences, Inc., Fremont, CA, USA).

#### 2.4.7. Statistical Analysis

All tests were performed in three independent experiments, and data are expressed as mean ± standard error of the mean (SEM). Statistical analyses were assessed using repeated measures ANOVA followed by Bonferroni or Dunnett post hoc tests as appropriate, employing Prism 9.0 software (GraphPad Software, Boston, MA, USA). A *p*-value < 0.05 was considered statistically significant.

## 3. Results and Discussion

### 3.1. Cytotoxicity Evaluation

The research began by determining the concentrations to be used in subsequent experiments aimed at evaluating the potential mutagenicity of the various nitazenes under investigation.

We assessed the cytotoxic effects of the compounds after 26 h of treatment, corresponding to approximately two replication cycles of TK6 cells. Cell viability was measured across the tested concentration range (12.5–100 µM), and the resulting data were normalized to those obtained from negative control (0 µM).

Cytotoxicity assessment is crucial, as outlined in OECD Guideline No. 487, which recommends that the highest concentration used in mutagenicity studies should not induce cytotoxicity exceeding 55 ± 5%, thereby ensuring a minimum cell viability of 45 ± 5% [9]. Exceeding this threshold results in inhibited cell proliferation or cell death, which does not allow the accurate determination of MNi. Indeed, cell replication and survival are required for damage fixation and MNi formation.

As illustrated in Figure 2, cell viability remained well above the OECD-recommended threshold (green line) across all tested concentrations, except for the highest concentration of clonitazene (100 µM), which resulted in a notable reduction. Although etonitazene induced a greater reduction in cell viability at 100 µM compared with the same concentration of isotonitazene and metonitazene, the corresponding *p*-value was higher. This difference likely reflects greater variability among etonitazene replicates, which reduced the statistical power of Dunnett’s post hoc comparison despite a stronger biological effect.

Moreover, in accordance with OECD Guideline No. 487, it is required to demonstrate that a significant proportion of the cells analyzed have completed at least 1.5–2 replication cycles during the treatment period. In line with the cytotoxicity assessment, the guideline also define a cytostasis threshold of 55 ± 5% and suggest calculating the RPD, which should remain at or above 45 ± 5% [9].

The results indicate that cytostasis remains within acceptable limits up to 50 µM for clonitazene, up to 75 µM for both etonitazene and metonitazene, and at all tested concentrations for isotonitazene (Figure 3).

Cell cycle arrest is a key parameter in genotoxicity assessment, as it reflects the cell’s ability to respond to genomic damage. In the presence of DNA damage, healthy cells typically slow down or halt replication to allow the activation of DNA repair mechanisms, thereby preventing the transmission of genomic errors to daughter cells [13].

### 3.2. Apoptosis Evaluation

Genetic damage can elicit a range of cellular responses, including apoptosis. Apoptosis was evaluated using double staining with Annexin V Alexa Fluor 488 and PI. The results showed that etonitazene induced a two-fold increase in apoptotic cells at 50 µM, while lower concentrations exhibited effects similar to the negative control (Figure 4B). Similarly, isotonitazene caused a two-fold increase in apoptosis only at the highest concentration tested (100 µM), with lower concentrations showing no significant effect (Figure 4C). Metonitazene showed apoptosis levels equivalent to the negative control at 12.5 and 25 µM, whereas a two-fold increase was observed at 50 µM (Figure 4D). In contrast, clonitazene did not induce apoptosis at any of the concentrations tested, with values comparable to those of the negative control (Figure 4A). This observation suggests that, in the presence of a genotoxic agent, cells may fail to activate the programmed cell death pathway, thereby allowing the transmission of genetic damage to subsequent generations [12,15]. Further studies are needed to confirm whether the absence of apoptosis is associated with deficiencies in DNA repair processes.

Concentrations associated with a two-fold or greater increase in apoptosis relative to the negative control were leave out from the MN frequency evaluation.

### 3.3. MNi Frequency Evaluation

The concentrations selected for the mutagenicity test were based on the findings of the cytotoxicity, cytostasis, and apoptosis assessments. Specifically, clonitazene and metonitazene were tested at 25 and 50 µM; etonitazene at 12.5 and 25 µM; and isotonitazene at 50 and 75 µM.

To assess the potential mutagenic effects, the MNi frequency was determined in negative controls, in cultures treated with different nitazenes under investigation, and in positive control treated with Vinb, a known mutagen. The MN test was performed by FCM using the protocol previously described by Lenzi et al. [13].

This approach offers several advantages over traditional optical microscopy. It enables the analysis of a large number of cells, thereby ensuring greater statistical robustness of the results. Moreover, it provides more accurate and objective data—particularly important when investigating weakly genotoxic compounds—by eliminating the subjective interpretation inherent to optical microscopy. Additional benefits include significantly reduced analysis time, lower reagent consumption and associated costs, and a simplified workflow due to the elimination of certain procedural steps. The method is also characterized by high reproducibility, thanks to the use of widely available fluorophores.

Using this method, we demonstrated that clonitazene and isotonitazene induced an increase in MNi frequency at the highest tested concentration, greater than a doubling compared to the negative control, while the results for etonitazene and metonitazene were comparable to those of the negative control. Consequently, only clonitazene and isotonitazene resulted as mutagenic, indicating their potential to elicit chromosomal aberrations (Figure 5).

As expected, treatment with the positive control Vinblastine (0.0077 μM) produced a significant increase in MNi frequency compared with the untreated control, corresponding to an approximately 4–5-fold increase across all experiments and within the historical laboratory range, thus confirming the responsiveness of the assay system.

At first glance, the graphs showing MNi frequency may appear to exhibit large error margins. However, this apparent variability reflects the high sensitivity and precision of FCM and the method itself, as well as the inherently probabilistic nature of genotoxic events—highlighting the importance of performing repeated analyses to ensure reliability.

Clonitazene and isotonitazene resulted as mutagenic, but not etonitazene and metonitazene. These findings demonstrate that different compounds within the same class of NPS can exert distinct cellular effects.

Several of our previous studies have shown similar trends in structurally related compounds, such as mexedrone, α-PVP (alpha-pyrrolidinopentiophenone), and α-PHP (alpha-pyrrolidinohexanophenone), which all belong to the cathinone synthetic class but do not exert the same effects on genetic material [16].

Another example comes from a study in which we demonstrated a different behaviour of the same molecule, the 4-Methyl-5-(4-methylphenyl)-4,5-dihydroxazol2-amine (4,4′-DMAR). In fact, this substance presents itself as two possible isomers that exhibit distinct genotoxic profiles: (±)cis-4,4′-DMAR induced a statistically significant increase in MNi frequency, whereas (±)trans-4,4′-DMAR did not, indicating a lack of genotoxic potential [17].

Regarding opioids’ genotoxicity, in current literature, two relevant studies can be highlighted: one on the hydromorphone impurity 2,2-bishydromorphone, which showed no mutagenic or clastogenic properties based on in silico QSAR predictions and in vitro Ames and micronucleus tests; and another on tramadol, which evaluated its potential human health risks, highlighting that it can induce a statistically significant increase in the MN frequency [18,19].

For NSOs, in one study, we assessed the mutagenic potential of three fentanyl analogues, demonstrating that fentanyl—the pharmaceutical progenitor of the class—was non-mutagen, in contrast to its illicit non-pharmaceutical analogues (acrylfentanyl, furanylfentanyl, and ocfentanyl), which showed significant mutagenic effects [11].

Furthermore, we evaluated the genotoxicity of brorphine and four of its analogues—orphine, fluorphine, chlorphine, and iodorphine. The results revealed a statistically significant increase in MNi frequency for fluorphine, chlorphine, and iodorphine, but not for brorphine and orphine, suggesting that the former compounds are capable of inducing chromosomal damage [12].

Regarding nitazenes, no publications are currently available in the literature addressing their genotoxic potential. The mutagenicity observed in our study underscores the hazardous potential of clonitazene and isotonitazene—not only in terms of acute toxicity, but also with respect to long-term exposure, which may contribute to the onset of chronic conditions such as cardiovascular, neurodegenerative disorders, cancer and heritable genetic diseases [20,21].

Mutagenic activity was observed for clonitazene and isotonitazene, whereas etonitazene and metonitazene showed no genotoxic effects. This further confirms that even minimal structural variations among molecules—such as the substitution of a single functional group—can alter physicochemical properties (e.g., electrophilicity and DNA reactivity), metabolic activation pathways, and interactions with cellular components, ultimately influencing their genotoxic potential [22,23,24].

In particular, the mutagenic effects observed may be attributable to the induction of oxidative stress. Accordingly, further investigations will be undertaken to quantify reactive oxygen species (ROS) levels potentially elevated following exposure to the tested compounds. This analysis will help determine whether oxidative stress constitutes a plausible mechanism underlying the genotoxic effects observed in two compounds but not in the others. Indeed, differential modulation of ROS levels may contribute to the observed variations in genotoxicity, as may the distinct chemical structures of the compounds under investigation. Future research should therefore aim to elucidate these mechanistic aspects in greater detail.

Nonetheless, it is important to emphasize that, while such considerations are of substantial scientific interest, from a toxicological and forensic standpoint the primary concern remains the determination of a compound’s genotoxic potential. This has already been established for two of the compounds examined.

For those that appear to be non-genotoxic, the next critical step will involve confirming or refuting this preliminary observation through assessments of potential gene mutation induction and comprehensive metabolite analyses. Indeed, another critical aspect to consider is the role of metabolites. The biotransformation of these compounds may produce secondary metabolites with toxicological profiles that differ substantially from the parent molecule. Therefore, genotoxicity assessment should not be limited to the original compound but must also include its major metabolites, under experimental conditions that closely mimic human metabolism.

In this context, OECD Guideline No. 487 recommends evaluating mutagenicity with an exogenous metabolic activation system—such as the S9 mix—to detect potential genotoxic effects of metabolites. This step is particularly important given the limited metabolic capacity of TK6 cells, which represents a major drawback. The inclusion of the S9 mix is thus essential to provide a more comprehensive toxicological profile of the tested substances.

Therefore, further studies are fundamental to investigate the genotoxicity of etonitazene and metonitazene in the presence of the S9 mix.

## 4. Conclusions

Our study demonstrated for the first time that clonitazene and isotonitazene exhibit genotoxic properties, as evidenced by their ability to induce chromosomal aberrations, whereas etonitazene and metonitazene do not. A more comprehensive assessment of their genotoxic potential could involve evaluating their capacity to induce point mutations—for instance, through the “Bacterial Reverse Mutation Test”—and examining the behaviour of their metabolites. Although such analyses would provide valuable additional genotoxicological insights, the positive outcome of the MN assay for clonitazene and isotonitazene has already underscored the critical aspect of this research: it has revealed an additional and concerning toxicological issue related to nitazenes.

Indeed, the demonstrated genotoxic effects warrant increased attention to the potential for serious long-term consequences, given the well-established role of genotoxicity in the onset of various neurodegenerative and chronic degenerative diseases.

## Figures and Tables

**Figure 1 jox-15-00203-f001:**
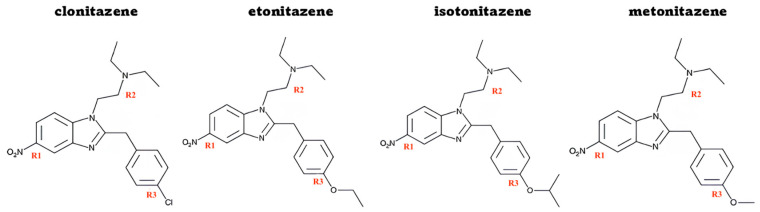
Chemical structure of nitazenes (clonitazene, etonitazene, isotonitazene and metonitazene) tested in the present study. These molecules differ by the substituent at position R3: clonitazene carries a chlorine atom, etonitazene an ethoxy group, isotonitazene an isopropoxy group, and metonitazene an isopropoxy group.

**Figure 2 jox-15-00203-f002:**

Viability of TK6 cells after 26 h treatment with (**A**) clonitazene, (**B**) etonitazene, (**C**) isotonitazene and (**D**) metonitazene at the concentrations reported compared to the untreated negative control (0 µM). Each bar represents the mean ± SEM of at least three independent experiments. Data were analyzed by Repeated Measures ANOVA followed by Dunnett post-test. The green line shows the OECD threshold. * *p* < 0.1; ** *p* < 0.01; *** *p* < 0.001; **** *p* < 0.0001.

**Figure 3 jox-15-00203-f003:**
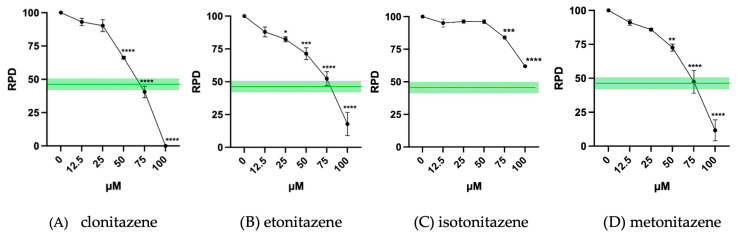
Relative Population Doubling (RPD) of TK6 cells after 26 h treatment with (**A**) clonitazene, (**B**) etonitazene, (**C**) isotonitazene and (**D**) metonitazene at the concentrations reported compared to the untreated negative control (0 µM). Each value represents the mean ± SEM of at least three independent experiments. Data were analyzed by Repeated Measures ANOVA followed by Dunnett post-test. The green line shows the OECD threshold. * *p* < 0.1; ** *p* < 0.01; *** *p* < 0.001; **** *p* < 0.0001.

**Figure 4 jox-15-00203-f004:**
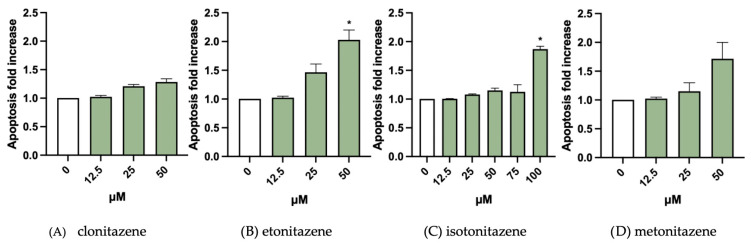
Apoptosis fold increase in TK6 cells after 26 h treatment with (**A**) clonitazene, (**B**) etonitazene, (**C**) isotonitazene and (**D**) metonitazene at the concentrations reported compared to the untreated negative control (0 µM). Each value represents the mean ± SEM of at least three independent experiments. Data were analyzed by Repeated Measures ANOVA followed by Bonferroni or Dunnett post-tests. * *p* < 0.1.

**Figure 5 jox-15-00203-f005:**
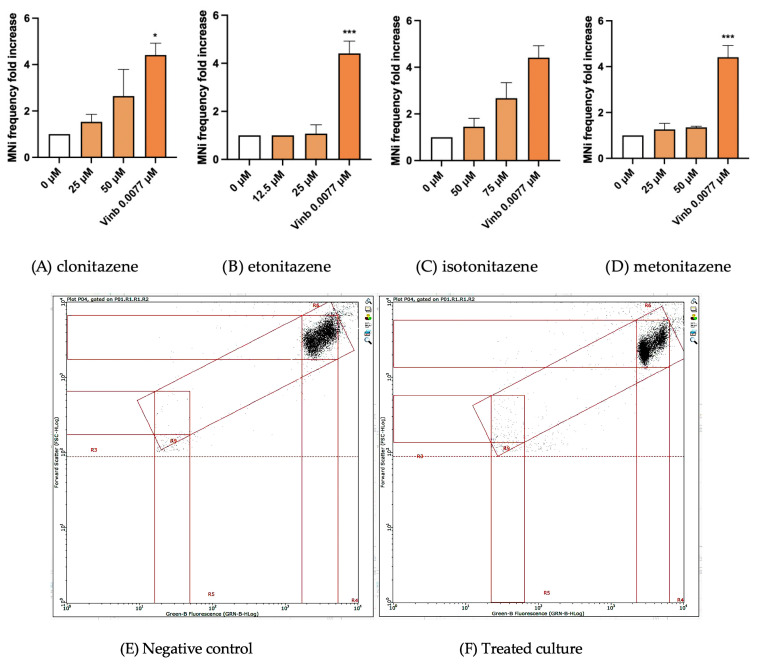
MNi frequency fold increase on TK6 cells after 26 h treatment with (**A**) clonitazene, (**B**) etonitazene, (**C**) isotonitazene and (**D**) metonitazene at the concentrations reported compared to the untreated negative control (0 µM) and positive control (Vinb). Each value represents the mean ± SEM of at least three independent experiments. Data were analyzed by Repeated Measures ANOVA followed by Bonferroni post-tests. * *p* < 0.1 vs. [0 µM]; *** *p* < 0.001 vs. [0 µM]. Representative FCM dot plots of nuclei and MNi scored in (**E**) a negative control and (**F**) in 50 μM clonitazene-treated cells.

**Table 1 jox-15-00203-t001:** Concentrations used for the measurement of MNi frequency.

	Selected Concentrations for MNi Test
clonitazene	25 µM and 50 µM
etonitazene	12.5 µM and 25 µM
isotonitazene	50 µM and 75 µM
metonitazene	25 µM and 50 µM

## Data Availability

The original contributions presented in the study are included in the article. Further inquiries can be directed to the corresponding author.

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
