# Peer review of "In Vitro Cytotoxic and Genotoxic Evaluation of Nitazenes, a Potent Class of New Synthetic Opioids"

_jox, 2025, doi:10.3390/jox15060203_

Round 1

Reviewer 1 Report

Comments and Suggestions for Authors

The manuscript titled “An Emerging Public Health Concern: Genotoxic Potential of Nitazenes, the Most Recent and Potent Classes of New Synthetic Opioids” presents a timely and relevant topic. It highlights the growing concern surrounding nitazenes, a new class of synthetic opioids with exceptionally high potency and toxicity compared to fentanyl and related compounds. The study draws attention not only to their addictive and toxic potential but also to their possible genotoxic effects, an underexplored area with important public health implications. Overall, the work provides valuable insight into an emerging toxicological issue that warrants further research.

I hope my observations and suggestions will be helpful in strengthening and improving your document.

I am attaching a document with my comments and suggestions.

Author Response

We thank the reviewer for the constructive feedback and careful reading of our manuscript.
We believe that the revisions made have significantly improved the quality and clarity of the work, and we hope that the revised version meets your expectations.

A point-by-point response to the reviewer’s comments is in the file uploaded. Please see the attachment.

Reviewer 2 Report

Comments and Suggestions for Authors

This is an interesting work entitled "An Emerging Public Health Concern: Genotoxic Potential of Nitazenes, the most recent and potent classes of New Synthetic Opioids", that is of merit to be published.

Author Response

We thank the reviewer very much for his appreciation of our work.

Reviewer 3 Report

Comments and Suggestions for Authors

Present the concentrations of all compounds using a single measurement scale (µM) to ensure comparability. Multiple and mass concentrations cannot be combined or compared.

It is necessary to clarify further what could cause the different mutagenic effects, given the very small structural difference in the tested compounds.

Figure 1: Indicate that structurally these four derivatives differ in the radical at R2; specify what R2 is and which radicals these are.

Line 135, 146, 167,179, 186, – specify the unit of measurement for cell concentration

Line 137, 147, – equalize

Concentrations used for testing should be shown in the table – refers to point 2.4.5 Measurement of MNi frequency

Link text into one paragraph lines 210 and 211

Uniform writing, i.e. replace h with hours in the entire text of the paper

Why are some words and references in the paper highlighted (colored) e.g. Line 156,186, 218,236,243,

Figure 2: It is unnecessary to write 0 µM and repeat; state that this is an untreated control and that the comparison was made in relation to control cells. Leave only the p-value limits and the indicated labels (asterisks).

In statistical analyses, state the chosen statistical significance limit.

The difference in survival when treated with 100 µM COMPOUND is small compared to other applied concentrations. Explain the difference in the p value, considering that with etonitazen, survival at the highest concentration is lower compared to the same concentrations of isotonitazen and metonitazen.

What about the SEM value in the control cells? Is it not determined when it is not visible on the graph? Is this valid for all graphs?

Figure 3: A, B, C, and D should be bold and aligned with Figure 2. RPD in the title should be written in full. It is unnecessary to write 0 µM repeatedly; instead, state that it is an untreated control and that comparisons were made relative to control cells.

Figure 4: Where are the SEM markings for the other concentrations? Are they established but not indicated? Edit the graph to add these. Level the scale at Apoptosis fold increase.

Figure 5: Adjust the height of the graphs to be equal. A, B, C, D, E, and F should be bold to be consistent with Figures 2 and 3. Unify the font size in the labels A–D and E–F. Improve the resolution of images 5E and 5F. Why are the results not shown for the positive control?

Line 338 – check the font size.

Why are some references coloured?

Is Lenzi reference number 11 or ?

Write the abbreviations PVP, PHP, and DMAR in full.

Author Response

(The authors gave the same response as above.)

Round 2

Reviewer 1 Report

Comments and Suggestions for Authors

The work meets expectations, I improved the presentation and writing, including the conclusion. 

 And the references are appropriate.